# Selection of Stable Reference Genes for Gene Expression Studies in Activated and Non-Activated PBMCs Under Normoxic and Hypoxic Conditions

**DOI:** 10.3390/ijms26146790

**Published:** 2025-07-15

**Authors:** Artur Wardaszka, Anna Smolarska, Piotr Bednarczyk, Joanna Katarzyna Bujak

**Affiliations:** 1Department of Physics and Biophysics, Institute of Biology, Warsaw University of Life Sciences, 02-787 Warsaw, Poland; artur_wardaszka@sggw.edu.pl; 2Center of Cellular Immunotherapies, Warsaw University of Life Sciences, 02-787 Warsaw, Poland; anna_smolarska@sggw.edu.pl

**Keywords:** PBMCs, hypoxia, reference genes, housekeeping genes, quantitative real-time PCR (qRT-PCR)

## Abstract

Immunotherapy has emerged as a key modality in cancer treatment, yet its effectiveness varies significantly among patients, often due to the metabolic stress imposed by the tumor microenvironment. Hypoxia, a major factor in the tumor microenvironment, results from the high metabolic rate of tumor cells and inadequate vascularization, impairing immune cells’ function and potentially influencing gene expression profiles. Despite the widespread use of quantitative real-time PCR in immunological studies, to the best of our knowledge, data on reference gene stability in human peripheral blood mononuclear cells under hypoxic conditions is limited. In our study, we assessed the expression stability of commonly used reference genes (*S18*, *HPRT*, *IPO8*, *RPL13A*, *SDHA*, *PPIA*, and *UBE2D2*) in both non-stimulated and CD3/CD28-activated peripheral blood mononuclear cells cultured under normoxic, hypoxic (1% O_2_), and chemically induced hypoxic conditions for 24 h. Analysis using four different algorithms—delta Ct, geNorm, NormFinder, and BestKeeper—identified *RPL13A*, *S18*, and *SDHA* as the most suitable reference genes for human peripheral blood mononuclear cells under hypoxic conditions. In contrast, *IPO8* and *PPIA* were found to be the least suitable housekeeping genes. The study provides essential insights into the stability of reference genes in peripheral blood mononuclear cells under hypoxic conditions, a critical but understudied aspect of immunological research. Given the significant impact of hypoxia on T cell metabolism and function in the tumor microenvironment, selecting reliable reference genes is crucial for accurate gene expression analysis. Our findings will be valuable for future studies investigating hypoxia-driven metabolic reprogramming in immune cells, ultimately contributing to a better understanding of T cell responses in cancer immunotherapy.

## 1. Introduction

Immunotherapy has emerged as a pivotal treatment modality for various cancer diseases, including melanoma, non-small cell lung cancer, stomach cancer, kidney cancer, and acute lymphoblastic leukemia [1,2,3,4,5]. There are several approaches to cancer immunotherapy, including monoclonal antibodies (mAbs), small-molecule drugs, adoptive cell therapy, oncolytic viruses, cancer vaccines, and chimeric antigen receptor T cell (CAR-T) therapy [3]. However, the effectiveness of immunotherapy varies among patients, often due to metabolic stress induced by the tumor microenvironment (TME), with hypoxia being a key contributing factor [6]. Some types of immunotherapy that are more sensitive to hypoxia include immune checkpoint inhibitors, CAR-T cells, cancer vaccines, and cytokine-based therapies. Hypoxia promotes an immunosuppressive tumor microenvironment (TME) through HIF-mediated mechanisms, such as upregulation of *PD-L1* and *VEGF* and recruitment of regulatory immune cells. Hypoxia also alters the metabolism and survival of neoplastic cells, enhancing their resistance to immune-mediated killing and limiting the effectiveness of both adaptive and innate immune response [6]. Hypoxia, defined as a reduction in partial oxygen pressure below 15 mmHg in tissues, is a key microenvironmental factor promoting tumor progression, immune escape, and therapy resistance [7,8,9]. In solid tumors, uncontrolled proliferation of cancer cells and abnormal vascularization contribute to the formation of hypoxic microregions, which activate hypoxia-inducible factors (*HIF*) that regulate genes involved in angiogenesis, metabolism, and survival [10,11,12]. Under such conditions, *HIF* upregulated not only in neoplastic cells but also in various stromal and immune cells within the TME, including T lymphocytes, tumor-associated macrophages, dendritic cells, and myeloid-derived suppressor cells [13]. These adaptations in the TME impair immune cell function and affect gene expression profiles. Accurate normalization of gene expression data under hypoxic conditions requires the identification of stable housekeeping genes. Several studies have identified reference genes in peripheral blood mononuclear cells (PBMC) derived from animals adapted to hypoxia [14,15] and in various types of human cells under hypoxic conditions, including alveolar epithelial cells, endothelial cells, renal cells, breast cancer cell lines, and melanoma [16,17,18,19]. However, studies focusing specifically on PBMC exposed to hypoxic conditions remain limited [20,21,22].

This study aimed to select a reliable and stable reference gene for studies on both non-activated and activated peripheral blood lymphocytes under hypoxic conditions (1% O_2_) and chemical hypoxia. The latter does not involve a reduction in oxygen levels but instead mimics hypoxic conditions by stabilizing *HIF-1α* under normoxic conditions. Chemical hypoxia was induced using cobalt chloride, a well-established hypoxia-mimetic agent known to stabilize *HIF-1α* by inhibiting its prolyl hydroxylation and subsequent degradation [23]. The stability of candidate housekeeping genes (*S18*, *HPRT*, *IPO8*, *RPL13A*, *SDHA*, *PPIA*, and *UBE2D2*) (Table 1), selected through a literature review [14,15,20,21,22,24], was evaluated in human PBMCs exposed to hypoxia using the Bryt^TM^. Green quantitative real-time PCR (qPCR) technique. Seven candidate reference genes were assessed for expression stability using geNorm, NormFinder, BestKeeper, and the comparative ΔCt method (difference in cycle threshold values). In addition, the web-based tool RefFinder was employed, which integrates all four algorithms to generate a comprehensive comparative ranking of candidate genes.

Identifying a stable reference gene tailored to hypoxic conditions in both non-activated and activated PBMCs is essential for ensuring the accuracy of gene expression analyses in immunological and cancer research. The findings of this study will support more reliable normalization in reverse transcription quantitative real-time PCR (RT-qPCR) experiments, contributing to a better understanding of how hypoxia influences immune cell function. This, in turn, may facilitate the development of more effective immunotherapeutic strategies in hypoxia-affected tumor environments.

## 2. Results

### 2.1. Standard Curve and PCR Efficiency

To assess amplification efficiency, a standard curve was generated for each primer pair using a 4-point, 2-fold serial dilution series of cDNA. From the combined data points, the slope of each standard curve was calculated. The resulting PCR efficiencies ranged from 91.13% to 113.6%, which falls within the acceptable range for qPCR analysis. Also, the standard curves generated for the candidate reference genes exhibited linear correlations, with values ranging from 0.992 to 0.999. Detailed efficiency values, correlation coefficients (R^2^) and amplicon sizes for each primer pair are presented in Table 2. Each primer pair was tested for specificity by melting curve analysis, which consistently revealed a single peak, indicative of a specific amplification product. Additionally, amplicons were separated on a 2% agarose gel stained with SYBR Safe, and single bands of the expected size were observed for all primer sets, further confirming amplification specificity (Appendix A).

### 2.2. Expression Profiles of Candidate Reference Genes

In our study, we analyzed the expression stability of seven candidate reference genes in human PBMCs, selected based on previous reports on reference gene validation [20,21,24]. Gene expression was assessed under three experimental conditions: normoxia, hypoxia (1% O_2_), and chemically induced hypoxia using CoCl_2_. Each of these conditions included both unstimulated and CD3/CD28-activated cell groups. The Ct values of all tested genes showed considerable variation, ranging from 17.10 to 33.86 across all samples and conditions. Among the analyzed genes, *RPL13A* exhibited the highest expression levels, with Ct values ranging from 17.10 to 21.63, while IPO8 showed the lowest expression, with Ct values between 24.82 to 33.86 across all samples (Figure 1). The standard deviation (SD) of Ct values ranged from 3.03 for *IPO8* to 1.11 for *SDHA*. Additionally, the raw Ct values of each gene were analyzed separately across all experimental conditions and for each individual donor (Appendix A). The Ct values varied depending on the gene and the treatment condition, with *RPL13A* and *S18* showing relatively stable expression across all treatment groups and donors. In contrast, deviations in Ct values among donors were noted for *IPO8* and *PPIA*.

The coefficient of variation (CV) for Ct values among all donors and treatments was lowest for SDHA, (4.56%), followed by RPL13A (6.26%), suggesting relatively consistent expression, whereas the genes PPIA (12.5%) and IPO8 (10.78%) displayed the highest variability (Appendix A).

### 2.3. Evaluation of Gene Expression Stability Using Delta Ct, NormFinder, BestKeeper, GeNorm, and RefFinder

To determine the most suitable reference genes for accurate normalization under our experimental conditions, we evaluated the expression stability of the seven candidates using four widely accepted algorithms: the ΔCt method, NormFinder, BestKeeper, and GeNorm. Additionally, we integrated the results using the RefFinder online tool to obtain a comprehensive ranking of gene stability.

Gene expression stability was assessed using the ΔCt method and SD, following the approach described by Silver et al. (2006) [27]. Briefly, we compared all possible gene combinations and analyzed the variation in ΔCt values across the samples. Among the tested genes, *RPL13A* exhibited the lowest deviation, with an average SD of 1.34. In contrast, IPO8 showed the greatest variability, with an SD of approximately 2.86 (Appendix A). The remaining genes demonstrated intermediate levels of expression stability. Based on the overall variability observed, the stability ranking of the reference genes is as follows: *RPL13A*, *S18*, *UBE2D2*, *SDHA*, *HPRT*, *PPIA*, and *IPO8* (Figure 2). These results indicate that *RPL13A* is the most suitable reference gene for normalization under the tested conditions.

To further evaluate the expression stability of candidate reference genes, we applied the NormFinder algorithm. The analysis was conducted across all experimental conditions, including normoxia, hypoxia, and chemically induced hypoxia, as well as within subgroups of non-stimulated and activated cells under each treatment condition. According to the NormFinder stability index, *RPL13* was identified as the most stable reference gene, whereas *IPO8* showed the lowest stability, with stability values of approximately 0.087 and 0.197, respectively (Figure 3a). These findings are consistent with the results obtained using the ΔCt method, which likewise identified *RPL13A* as the most stable and *IPO8* as the least stable reference gene.

In the next step of our analysis, we employed the geNorm algorithm, which defines the M value as a measure of gene expression stability. The M value represents the average pairwise variation of a given candidate reference gene relative to all other tested genes. A lower M value indicates more stable expression, while higher values reflect greater variability [28,29]. In our study, M values were calculated for all candidate genes across the entire set of samples. The results showed that *UBE2D2* and *S18* had the lowest M value of 0.38, indicating the highest expression stability, whereas *IPO8* exhibited the highest M value of 1.89, suggesting that it is the least stable gene among those analyzed (Figure 3b). Based on the geNorm analysis, the default threshold for acceptable gene stability is an M value below 1.5; therefore, genes with stability values exceeding 1.5 such as *IPO8* and *PPIA* need to be excluded from further gene expression analyses. To determine the optimal number of reference genes for accurate normalization, we performed pairwise variation analysis using geNorm. The lowest pairwise variation was observed for V3/4 = 0.142, which falls below the commonly accepted cut-off value of 0.15, indicating that the use of three reference genes is sufficient for reliable normalization in our experimental setup (Appendix A). The results obtained from the geNorm analysis are partially consistent with those generated using the ΔCt and NormFinder methods. In all approaches, *IPO8* consistently emerged as the least stable reference gene, while *RPL13A*, *UBE2D2*, and *S18* were identified among the most stable candidates across the experimental conditions.

Subsequently, we compared the data using the BestKeeper algorithm, which evaluates gene expression stability based on descriptive statistics, including the SD and CV across all samples. The most stable reference genes were selected based on the lowest SD values, with *SDHA* and *RPL13A* showing the highest stability, having SD values of 0.83 and 0.97, respectively. Also, among the tested genes, *RPL13A* showed the highest correlation coefficient (*r* = 0.922), suggesting a strong association with the BestKeeper index and supporting its suitability as a stable reference gene (Table 3; Appendix A). In contrast, the least stable gene was *IPO8*, with an SD of 2.76, which is in accordance with other methods (Table 3).

Finally, we used the web-based tool RefFinder, which integrates the results from all previously applied algorithms (ΔCt method, NormFinder, geNorm, and BestKeeper) to calculate a comprehensive stability ranking of candidate reference genes (Appendix A). Based on the combined analysis, the most stable gene identified was *RPL13A*, followed by *S18* and *SDHA*, while *IPO8* was ranked as the least stable reference gene across all tested conditions (Figure 4).

We also used RefFinder to assess the most stable reference genes within each experimental condition separately—hypoxia, normoxia, and chemical hypoxia (Appendix A). This analysis revealed condition-specific differences in gene stability rankings, with *RPL13A* and *UBE2D2* showing the highest stability in hypoxia, *RPL13A* and *UBE2D2* in normoxia, and *RPL13A* and *S18* under chemical hypoxia. These findings highlight the importance of validating reference genes in a context-dependent manner.

Overall, among all candidate reference genes tested, *RPL13A*, *UBE2D2*, and *S18* consistently demonstrated the highest expression stability across all experimental conditions (Table 4). In contrast, *IPO8* repeatedly appeared as the least stable gene in all analyses and should therefore be avoided as a reference gene for gene expression studies in PBMCs under the tested conditions.

### 2.4. Validation of Selected Reference Genes

To validate the suitability of the selected reference genes identified in our analyses, we examined the relative expression levels of the *HIF-1α* gene using different normalization strategies. Specifically, we assessed the performance of *RPL13A* and *SDHA* individually, as well as in combination with *RPS18* (arithmetic and geometric means of Ct values), and also included a combination of three genes identified by geNorm (*RPL13A*, *RPS18*, and *UBE2D2*). For comparison, we also tested *IPO8*, identified in our study as one of the least stable genes (Figure 5). The relative expression was calculated using the extended ΔCt method and normalization to the average Ct of multiple reference genes, as previously described by Riedel et al., (2014) [30].

Our results showed that normalization using the arithmetic or geometric mean of *RPL13A*, *SDHA*, and *RPS18* yielded consistent and biologically plausible patterns of *HIF-1α* expression, including a statistically significant increase in activated T cells under normoxic conditions (Figure 5d,e). Similarly, normalization using either *SDHA* or *RPL13A* alone produced a comparable expression pattern (Figure 5a,b). The combination of the three most stable genes according to geNorm (*RPL13A*, *RPS18*, and *UBE2D2*) also gave similar results (Figure 5f). In contrast, normalization using IPO8 led to a markedly divergent expression profile of *HIF-1α*, with substantial variability across samples (Figure 5c). This discrepancy may be attributed to the high variability in *IPO8* Ct values observed in our dataset, further confirming its unsuitability as a reference gene under our experimental conditions.

## 3. Discussion

Quantitative real-time polymerase chain reaction is a highly sensitive and reproducible method for quantifying gene expression, which is the foundation of many areas of biological sciences, including the analysis of signaling pathways [31,32], the diagnosis of infectious diseases such as COVID-19 [33], and the identification of tumor markers [34]. However, the reliability of qPCR results is influenced by several factors, with the selection of an appropriate reference gene being one of the most critical. An ideal reference gene is characterized by stable transcription across different cell types and tissues regardless of the experimental conditions. Despite the common use of genes such as *ACTB*, *S18*, and *GAPDH* as reference genes, numerous studies have demonstrated that their expression levels may vary depending on cell type, developmental stage, and experimental conditions [35,36,37]. Thus, validation and selection of adequate reference genes are essential to ensure the accuracy and reliability of gene expression analyses using qPCR. Importantly, a limitation of our study is the relatively small number of donors (*n* = 5), all of whom were male. While this design ensured a certain degree of uniformity, it does not capture the full spectrum of biological variation present in the general population, including potential sex-related differences in gene expression and immune responses. Therefore, the findings should be validated in larger, more diverse cohorts.

In the present study, we focused on PBMCs, which were cultured under conditions of normoxia, hypoxia, and chemically induced hypoxia using CoCl_2_. PBMCs represent a heterogeneous population of immune cells, including mostly lymphocytes and monocytes, and serve as a readily available source of genetic material that reflects the gene expression profile of individuals [38,39]. This makes them a valuable tool for investigating differential gene expression, particularly in translational and clinical research. Due to their central role in immune responses, PBMCs are of high relevance in studies related to protection against pathogens, the pathophysiology of autoimmune diseases, and anti-tumor immunity. Importantly, in the context of cancer and the growing field of immunotherapy, PBMCs offer an accessible model to explore immune cell behavior under pathological conditions, including hypoxia, which is a hallmark of the TME [40,41]. Given the potential of PBMCs to provide insight into how immune cells respond to various stimuli—including hypoxic stress—it was essential to identify reliable reference genes suitable for normalization in RT-qPCR studies. Accurate normalization is critical for interpreting gene expression changes that may indicate whether cellular responses are adaptive or detrimental to PBMC function and physiology. RT-qPCR results, when properly normalized, can offer a glimpse into the molecular mechanisms underlying immune responses and contribute to our understanding of how these cells behave under both physiological and disease-related conditions.

In the conducted study, the stability of seven potential reference genes was evaluated under normoxia, hypoxia, and chemical hypoxia conditions. We selected the candidate reference genes based on previous publications that identified them as the most stable or suitable for normalization in studies involving PBMCs and/or hypoxic conditions [20,24,42,43]. The genes used in our study, namely *RPL13*, *IPO8*, *S18*, *SDHA*, *HPRT*, *UBE2D2*, and *PPIA*, were also selected based on the assumption that they are not co-regulated, which is a crucial criterion for reliable normalization in RT-qPCR analysis [44,45,46,47,48,49,50]. This helps to ensure that observed expression stability is not a result of shared regulatory mechanisms but reflects true independence across experimental conditions.

For the purpose of identifying suitable reference genes, several algorithms have been developed to date, including ΔCt, BestKeeper, NormFinder, geNorm, and the online tool RefFinder. Each of these methods has its own advantages and limitations, and as a result, there is no clear consensus on the single best approach for selecting optimal reference genes.

The ΔCt method is based on comparing the relative expression of pairs of genes within each sample. It assumes that the ΔCt value between two genes remains constant across different samples if the genes are stably expressed or co-regulated. This method is simple and straightforward, which is one of its main advantages. Another strength is its suitability for working with limited or hard-to-quantify biological material. This method was originally described by Silver et al. (2006) [27] in the context of gene expression studies in reticulocytes. The authors emphasized that in this particular cell type, RNA yield is often low, and accurate quantification may not be feasible, making the ΔCt method particularly useful under such constraints.

BestKeeper is an Excel-based tool that evaluates the expression stability of candidate reference genes using raw Ct values. It calculates descriptive statistics, including SD and CV, to assess gene expression variability. According to this method, genes with an SD greater than 1 are considered inconsistent, while those with an SD below 1 are regarded as stably expressed [51]. BestKeeper also allows for the inclusion of primer efficiency in the analysis. However, the tool does not directly rank genes from most to least stable. Instead, it provides statistical output tables from which users can infer the most suitable reference genes for normalization.

NormFinder and geNorm both operate on relative quantities derived from Ct values. NormFinder estimates the overall expression variation of candidate reference genes and additionally provides separate measures of intra- and intergroup variability. A notable advantage of this algorithm is its reduced susceptibility to bias introduced by co-regulated genes, making it a robust tool for identifying stable reference genes across heterogeneous sample groups [52]. The geNorm algorithm, in turn, assesses the stability of candidate reference genes by calculating pairwise variation, expressed as an M value—defined as the average pairwise variation of a given gene with all other candidate genes, where a lower M value indicates higher expression stability. In addition to ranking genes, GeNorm determines the optimal number of reference genes required for accurate normalization by computing a normalization factor, defined as the geometric mean of the selected reference genes. This approach is based on the premise that relying on a single reference gene may introduce normalization errors [49].

RefFinder is an online tool that integrates the results from four widely used algorithms, namely ΔCt, NormFinder, BestKeeper, and GeNorm, and provides a comprehensive ranking of candidate reference genes. It offers a straightforward and user-friendly interface; however, it does not account for individual primer efficiencies during analysis.

In our study, *IPO8* consistently ranked as the least stable reference gene across all applied algorithms, including the ΔCt method, GeNorm, NormFinder, and BestKeeper. The comprehensive ranking provided by RefFinder also indicated that *IPO8* is the least suitable reference gene for our experimental conditions. Interestingly, *IPO8* has previously been reported as one of the most stable reference genes in clinical lung specimens and human adipose tissues [53,54]. To address the observed discrepancy, it is important to note that both NormFinder and GeNorm assign lower M values to genes with higher expression stability. Generally, an M value below 0.15 in NormFinder and below 1.5 in GeNorm is considered acceptable. In the study by Nguewa et al. [54], IPO8 was reported as best reference gene in clinicopathological lung specimens; however, in in vitro-cultured lung cell lines, IPO8 had an M value of 0.758 in NormFinder and 0.852 in GeNorm, placing it among the least stable reference genes in that context. However, Rácz et al. (2021) [55] demonstrated that *IPO8* is among the recommended reference genes for gene expression studies in various cell lines, including both normal and cancer-derived cells based on GeNorm, NormFinder, ΔCt, and BestKeeper. In normal cell lines, IPO8 showed favorable stability with M values of 0.381 in GeNorm, 0.205 in NormFinder, and 0.55 in the ΔCt method. In cancer-derived cell lines, the corresponding values were 0.346, 0.212, and 0.535, respectively. It should be emphasized that these results were obtained using established cell lines, which consist of clonal populations and thus provide high technical reproducibility across replicates. In contrast, our study used primary PBMCs, which represent a heterogeneous mixture of immune cell types that can vary significantly between donors. This biological variability may contribute to the observed differences in gene expression stability. Interestingly, *IPO8* along with *RPL13A*, *TPP*, and *SDHA* were identified as suitable reference genes in CD3/CD28-activated T cells [24]. On the other hand, a recently published study by Tóth et al. (2025) [56] demonstrated that *IPO8*, along with *PPIA*, were among the least stable genes for use as reference genes in cell cycle-dependent gene expression analysis in two human leukemia cell lines: U937 (monocytic) and MOLT4 (T cell). Notably, both of these cell types (monocytes and T cells) are also present in PBMCs, which constitute the cellular model used in our study. Moreover, our experimental setup included activated T cells, which are expected to undergo proliferation and enter the cell cycle. Taking into account the cell type and the activation status, our results are in agreement with the findings of Tóth et al. (2025) [56]—in our analysis, *PPIA* is also ranked as one of the least suitable reference genes, after *IPO8*. However, based on previous findings, *PPIA* has been identified as one of the most suitable reference genes in PBMCs from obese asthmatic subjects [57]. Similarly, Cinar et al. (2013) [58] reported the geometric mean of the *PPIA* and two other genes (*RPL4* and *B2M*) as suitable for normalization in porcine PBMC upon bacterial infection, while in non-activated PBMCs, *PPIA*, *B2M*, and *GAPDH* were proposed. These findings indicate that *PPIA* is, in certain conditions, suggested as a potential reference gene, often used in combination with others, depending on the specific experimental context and cellular stimulation status. Interestingly, in our study, in both IPO8 and PPIA, some inter-donor variation in Ct values was observed, which might influence the results. Such variability is an inherent feature of biological samples derived from primary human cells and reflects the natural diversity of physiological states across individuals. Donors may differ in their general physiology, immune history, and current physiological status, including subclinical or undiagnosed inflammatory or metabolic conditions. Such factors can influence the composition and activation state of immune cell populations, potentially affecting gene expression patterns. This biological heterogeneity is an expected characteristic of primary human samples and may contribute to the observed inter-donor variation in transcript levels. Rather than being a limitation, this variation highlights the importance of validating reference genes under realistic, variable biological conditions.

In addition to *IPO8* and *PPIA*, our results indicated that *HPRT* might not be the best reference gene under the experimental conditions used in this study. *HPRT* is a commonly employed housekeeping gene and has been widely used as a reference gene across various cell types, organisms, and experimental settings. Indeed, numerous studies have reported its stable expression; for example, Liu et al. (2023) [59] identified *HPRT* as an ideal reference gene in rat dorsal root ganglion (DRG) neurons. Similarly, *HPRT*, together with *RPL27*, was among the most stable genes in the mouse choroid plexus [60]. However, other studies have also reported, perhaps surprisingly, that *HPRT* is not appropriate as a reference gene in cancer-related studies due to its highly variable expression [61]. In the study by Verma et al., *HPRT* was identified as one of the more stable reference genes in cattle PBMCs from the hot arid normoxia group [14]. However, in the cold arid hypoxia group, *HPRT* fell within the least stable reference genes, and it was likewise excluded from the panel of stable genes in the combined analysis of both conditions. Notably, these investigations were conducted on bovine PBMCs under hypoxic stress—an experimental design analogous to our study of human PBMCs exposed to hypoxia. The divergent stability profiles observed for *HPRT* across species and oxygen conditions further underscore the critical importance of validating reference genes in each specific biological context.

Our results indicate that *RPL13A* is the most stable reference gene under our experimental conditions, as consistently demonstrated by the ΔCt method, NormFinder analysis, and BestKeeper data. Following *RPL13A*, the genes *RPS18*, *SDHA*, and *UBE2D2* ranked as the next most stable candidates. The convergence of results across multiple algorithms supports the reliability of *RPL13A* as a robust reference gene for normalization in our human PBMC model, particularly under hypoxic conditions. Studies conducted in other species, such as turkey (*Meleagris gallopavo*), have also indicated that *RPL13A* may serve as a suitable reference gene for normalization in central immune organs, including the bursa of Fabricius, thymus, and spleen [62]. *RPL13A* has also previously been identified as a reliable reference gene in human CD3/CD28-stimulated T lymphocytes, which is in line with our findings [24]. In contrast, other studies investigating T cells and PBMCs infected with the influenza virus demonstrated that *RPL13A*, along with *GAPDH*, were among the least reliable reference genes [20]. In paper by Roy et al. [20], the stability values for RPL13A were geNorm M = 0.556, NormFinder stability = 0.479, and BestKeeper SD = 0.52, while in a paper by Ledderose et al. [24], which employed CD3/CD28-stimulated T lymphocytes, as in our study, RPL13A showed much better performance, with geNorm M = 0.793, NormFinder = 0.083, and BestKeeper SD = 0.40. These differences might be related to the variation in experimental conditions: in the study by Roy et al. [20], the PBMCs were frozen, thawed, and then infected with influenza virus for 8 h, while in the study by Ledderose et al. [24], freshly isolated lymphocytes were stimulated for 24 h with CD3/CD28. There are also notable differences in Ct value patterns. In study by Roy et al. [20], RPL13A appeared to have relatively low expression, with Ct values around 30 cycles—higher than genes such as RPS18, UBE2D2, or SDHA. In our study, however, RPL13A showed lower Ct values than SDHA, which is consistent with the data from paper by Ledderose et al. [24], where a similar Ct pattern was observed in CD3/CD28-stimulated T cells. While it is difficult to fully explain the observed discrepancies between studies, factors such as RNA quality, stimulation time, and cell viability after thawing may contribute. Nevertheless, Ledderose et al. [24] recommended RPL13A and SDHA as suitable reference genes, which aligns with our results from NormFinder and BestKeeper analyses. On the other hand, Roy et al. [20] suggested UBE2D2 and RPS18 as the most stable reference genes, which matches the top-ranked pair identified in our geNorm analysis. This further underscores the need for context-specific validation of reference genes.

In numerous publications involving human T lymphocytes or PBMCs as the study material, *S18* is frequently identified as a reliable reference gene [20,42,63]. Similarly, in bottlenose dolphins (*Tursiops truncatus*), *S18* was identified by geNorm as one of the most stable reference genes, together with *RPL18*, for normalization in gene expression studies using blood samples. However, Baddela et al. [64] reported that in the case of cultured bovine granulosa cells, expression of *S18*, along with *RPLP0*, was inconsistent, especially under hypoxia and high cell plating density. This observation suggests that caution should be exercised when using *S18* as a reference gene under hypoxic conditions, as its expression may be sensitive to specific cellular stressors or environmental factors.

In addition to *S18*, two other genes—*SDHA* and *UBE2D2*—also emerged as promising candidates for reference gene selection in our study. According to NormFinder analysis, *SDHA* was ranked as the second most stable gene after *RPL13A*. Furthermore, BestKeeper analysis identified *SDHA* as the top candidate, based on its lowest standard deviation (SD = 0.83) and coefficient of variation (CV = 3.44), indicating expression consistency across samples. In porcine T cells, both with and without lipopolysaccharide (LPS) stimulation, *SDHA*, together with *RPL19*, was recommended as a suitable reference gene for T lymphocytes based on geNorm analysis [65]. Similarly, in human CD8^+^ T cells, *SDHA*, along with *18S* rRNA, was identified as one of the most stable reference genes upon cell activation [42]. Moreover, other studies have shown that *SDHA*, in addition to being a reliable reference gene for T cells, also appears among the most stable genes in untreated total blood leukocytes [24]. In our study, *SDHA* ranked highly across multiple stability analyses, which may be attributed to the fact that the majority of cells in our experimental samples were T lymphocytes. This cellular composition could contribute to the consistent expression of *SDHA*, aligning with previous reports that identified this gene as stable specifically in T cell populations. *UBE2D2*, on the other hand, was highlighted as one of the most stable genes alongside *S18* by the geNorm algorithm. In the ΔCt-based ranking, *UBE2D2* was placed third, immediately after *RPL13A* and *S18*. Similarly to our findings, Roy et al. reported that *UBE2D2* and *S18* were consistently among the most stable reference genes in both PBMCs and T cells, as determined by the comparative ΔCt method and validated across multiple algorithms, including BestKeeper, NormFinder, and geNorm [20]. Taken together, the results of our analysis suggest that both *SDHA* and *UBE2D2* may serve as reliable options for normalization, particularly when used in combination with other validated genes.

As a good practice, using multiple validated reference genes is recommended, as it significantly enhances the robustness of normalization [66]. Among the algorithms used for reference gene evaluation, geNorm uniquely provides information on the optimal number of reference genes required for accurate normalization under given experimental conditions. The optimal number of reference genes in geNorm is determined by calculating the pairwise variation (V) between sequential normalization factors (NFn and NFn+1). When the Vn/n+1 value falls below the threshold of 0.15, adding more genes is no longer necessary. In our study, three reference genes were required, as the V3/4 value (0.142) was the first to drop below the 0.15 cut-off.

Importantly, it is not advisable to directly adopt reference genes from published studies without first conducting a pilot validation under one’s specific experimental conditions. Gene expression is a highly dynamic parameter influenced by numerous variables, many of which are difficult to predict. Therefore, it is advisable to validate candidate reference genes under specific experimental conditions and determine the optimal number needed for accurate normalization.

It is also important to note that different algorithms used in reference gene validation may yield varying results, as they rely on distinct statistical approaches to assess gene expression stability. Despite these methodological differences, in our study, all four algorithms along with RefFider ranking generally pointed to a similar group of genes as the most and least suitable for normalization under our specific experimental conditions. It is also worth emphasizing that discrepancies in gene ranking across studies may stem from the common practice of neglecting primer efficiency correction. As highlighted in the publication by Spiegelaere et al. [67], analyses based on raw Cq values without correcting for individual primer efficiencies may produce results that differ substantially from those based on efficiency-corrected data, ultimately affecting the identification of the most stably expressed genes. RefFinder is a convenient and fast tool—freely available online—which makes it an attractive option for initial evaluation. However, it should be noted that this tool relies on raw Ct values, without correcting primer efficiency. As a result, its outcomes may not always align with the results obtained from other algorithms that account for amplification efficiency. For this reason, it is always recommended to use multiple software tools when assessing reference gene stability in order to obtain a more robust and reliable normalization strategy.

To further assess the validity of the selected reference genes, we analyzed the relative expression of *HIF-1α* under different experimental conditions using several normalization strategies. Our results confirmed that normalization using the arithmetic or geometric mean of *RPL13A*, *SDHA*, and *RPS18* provided consistent and biologically relevant data, with a significant upregulation of *HIF-1α* observed in normoxia-activated T cells. In contrast, normalization with *IPO8*—identified as one of the least stable genes—resulted in highly variable and divergent expression profiles, likely due to its substantial Ct value variability in our dataset. Interestingly, we observed an increase in *HIF-1α* mRNA in activated PBMCs under normoxic conditions, while hypoxia alone (1% O_2_) did not induce its expression to the same extent. This finding is in line with previous studies demonstrating that *HIF-1α* expression in T cells is not solely dependent on low oxygen availability. Instead, TCR stimulation under normoxia, through activation of NF-κB and NFAT pathways, has been shown to transcriptionally upregulate *HIF-1α* [68]. Additionally, proinflammatory cytokines such as TNF-α and IL-1β may further enhance *HIF-1α* expression in a hypoxia-independent manner via NF-κB activation [69,70]. *HIF-1α* plays a pivotal role in regulating T cell responses, including the metabolic reprogramming towards glycolysis and shaping effector functions such as IFN-γ production [71,72]. Moreover, its regulatory influence on Th17/Treg balance via transcriptional activation of RORγt and degradation of FoxP3 further emphasizes its immunomodulatory potential [73].

## 4. Materials and Methods

### 4.1. PBMC Isolation

Healthy human donor PBMCs were isolated from buffy coats provided by the Centre for Blood Donation and Hemotherapy of the Ministry of Interior and Administration of Republic of Poland in Warsaw (CKiK MSWiA, Warsaw, Poland). Information about the donors from whom buffy coats were obtained is included in Table 5. The isolation process commenced within four hours of blood collection and was conducted under sterile conditions following the SepMate™ isolation protocol. Briefly, blood samples were diluted 1:1 with phosphate-buffered saline (VWR^®^, phosphate-buffered saline (PBS), pH 7.4, Avantor, Radnor, PA, USA) supplemented with 2% Fetal bovine serum (FBS) (Fetal Bovine Serum, EURX, Gdansk, Poland) at room temperature (RT) and carefully layered onto Histopaque-1077 (Histopaque^®^-1077 Hybri-Max™, Sigma-Aldrich, Merck KGAA, Darmstadt, Germany) in 50 mL SepMate™ tubes (SepMate™-50 (IVD), STEMCELL Technologies Canada Inc., Vancouver, BC, Canada). Centrifugation was performed at 1200× *g* for 10 min with the brake engaged at RT. Following centrifugation, the leukocyte layer was transferred to new tubes and washed twice with PBS + 2% FBS, followed by additional centrifugation at 300× *g* for 8 min at RT. To remove residual erythrocytes, the resulting cell pellet was treated with red blood cell lysis buffer (eBioscience™ 1X RBC Lysis Buffer, Invitrogen™, ThermoFisher Scientific, Waltham, MA, USA) according to the provided protocol. Cell viability was assessed using trypan blue staining.

Our analysis was limited to peripheral blood mononuclear cells (PBMCs), which include T cells, B cells, NK cells, and monocytes. Granulocytes, erythrocytes, and platelets were excluded during the isolation procedure.

### 4.2. Cell Culture

PBMCs were cultured in complete medium—RPMI 1640 (Cell culture medium—RPMI 1640, Biowest, Avantor, Radnor, PA, USA) supplemented with 10% FBS, non-essential amino acids (MEM Non-Essential Amino Acids 100×, GibcoTM, ThermoFisher Scientific, Waltham, MA, USA), 1 mM sodium pyruvate (Sodium Pyruvate (100 mM), GibcoTM, ThermoFisher Scientific, Waltham, MA, USA), 100 U/mL penicillin (Penicillin: Streptomycin solution 0.06/0.1 g/L, Biowest, Avantor, Radnor, PA, USA), 100 μg/mL streptomycin (penicillin: streptomycin solution 0.06/0.1 g/L, Biowest, Avantor, Radnor, PA, USA).

PBMCs were cultured at a density of 1 × 10^6^ cells/mL in complete medium at 37 °C under three distinct conditions: normoxia (5% CO_2_, atmospheric O_2_), hypoxia (≤1% O_2_), and chemical hypoxia induced by (500 µM) cobalt chloride (CoCl_2_) (Cobalt chloride 0.1 M solution, Sigma-Aldrich, Merck KGAA, Darmstadt, Germany) treatment under normoxic conditions for 24 h. Within each condition, cells were either left non-stimulated (NS) or activated using ImmunoCult™ Human CD3/CD28 T Cell Activator (ImmunoCult™ Human CD3/CD28 T Cell Activator, STEMCELL Technologies Canada Inc., Vancouver, BC, Canada) to induce T cell activation. This approach reflects the physiological relevance of T cells as the predominant lymphocyte population within PBMCs.

### 4.3. RNA Isolation and cDNA Synthesis

Total RNA was isolated from cells using the RNAqueous™-Micro Total RNA Isolation Kit (RNAqueous™-Micro Total RNA Isolation Kit, Invitrogen™, ThermoFisher Scientific, Waltham, MA, USA) and digested with DNase I included in the kit, according to the manufacturer’s protocol. RNA (200 ng) was transcribed into cDNA using the commercially available High-Capacity cDNA Reverse Transcription Kit (High-Capacity cDNA Reverse Transcription Kit, Applied Biosystems™, ThermoFisher Scientific, Waltham, MA, USA). The reaction conditions were as follows: primer annealing and initiation (10 min at 25 °C), cDNA synthesis (120 min at 37 °C), and enzyme inactivation (5 min at 85 °C). The cDNA was stored at −20 °C until further use in experiments.

### 4.4. Primer Design and Quantitative PCR with SYBR Green

Primers were either taken from publications or designed de novo using National Centre for Biotechnology Information (NCBI) software Primer Blast (version 4.1.0) [74] (Table 2). Primers were synthesized by the Institute of Biochemistry and Biophysics, Polish Academy of Sciences, via their Oligo platform for custom oligonucleotide synthesis. For each primer pair, a standard curve was generated using four 2-fold serial dilutions of sample cDNA. Primer specificity was confirmed by melting curve analysis and further validated by agarose gel electrophoresis. Based on the obtained standard curves, the slope and amplification efficiency calculated as E% = (10(−1/slope) −1) × 100% [51] were determined (Table 2).

qPCR was performed in 96-well plates using GoTaq^®^ qPCR Master Mix (GoTaq(R) qPCR Master Mix, Promega Corporation, WI, USA). Each reaction consisted of 10 µL containing 1 µL of cDNA, 1 µM of each primer (1 µL), 5 µL of Master Mix, and 3 µL of RNase free water (total volume 10 µL). The real-time qPCR was run on AriaDx Real-Time PCR System Agilent Technologies. The cycling conditions were 1 cycle of denaturation at 95 °C/2 min, followed by 40 three-segment cycles of amplification (95 °C/15 s, 55 °C/15 s, 60 °C/1 min) where the fluorescence was automatically measured during PCR, and product melting (95 °C/1 min, 55 °C to 95 °C/30 s in 0.5 °C intervals). The Ct value for each reaction was determined using the baseline correction method provided by the AriaDx Real-Time PCR System Agilent Technologies software (version 4.1.0). A melting curve analysis was performed for each primer pair to confirm the presence of a single gene-specific peak and the absence of primer-dimer formation.

PCR products were visualized using SYBR^TM^ Safe (SYBR™ Safe DNA Gel Stain, Invitrogen™, ThermoFisher Scientific, Waltham, MA, USA) on 2% agarose gel electrophoresis.

### 4.5. Data Analysis

The expression stability of the selected reference genes was assessed using 4 different algorithms—geNorm, NormFinder, BestKeeper, and the comparative ΔCt method—to provide a comprehensive ranking of gene expression stability. Each algorithm employs a distinct statistical approach: GeNorm calculates the average pairwise variation of given gene with all other candidate reference genes [49]; NormFinder estimates both intra- and inter-group variation to identify the most stable gene [52]. Values used in GeNorm and NormFinder were first converted to relative expression levels on a linear scale. This transformation was necessary, as both algorithms require input data that reflect linearized expression levels rather than logarithmic Ct values. Relative expression values were calculated using the ΔCt method by referencing each gene to a sample with the lowest Ct value and then linearized using the formula relative expression = E^ΔCt^, where E represents the amplification efficiency of the respective gene. BestKeeper uses raw Ct values to compute the SD and CV [51], and the comparative ΔCt method evaluates the relative expression of gene pairs within each sample [27]. Moreover, the RefFinder tool [75], a web-based platform that integrates several established algorithms (geNorm, NormFinder, BestKeeper, and the comparative ΔCt method), was employed to provide a comprehensive ranking of candidate reference genes. RefFinder assigns a weight to each gene based on its ranking across the individual algorithms and calculates the geometric mean of these weights to determine the overall stability ranking.

To assess the overall variability of each reference gene across all experimental conditions, we calculated the CV as the ratio of the standard deviation to the mean of Ct values, expressed as a percentage. Lower CV values indicate higher expression stability. This analysis was performed with all samples evaluated.

Graphs and data visualizations were generated using GraphPad Prism 8.0 software (GraphPad Software, San Diego, CA, USA).

## 5. Conclusions

In summary, hypoxia significantly influences immune cell function and gene expression, posing a major challenge for accurate molecular analyses in immunological and cancer research. Despite extensive studies on reference gene stability in various cell types and species, little attention has been given to human PBMCs under hypoxic conditions. This study addresses this gap by systematically evaluating the stability of seven candidate housekeeping genes in both non-activated and activated human PBMCs exposed to hypoxia. Identifying a reliable and stable reference gene under these conditions is essential for ensuring accurate normalization in qPCR experiments, ultimately contributing to more robust and reproducible insights into immune responses within hypoxic TME.

It is important to note that the different algorithms used for reference gene validation—such as BestKeeper, NormFinder, ΔCt method, geNorm, and RefFinder—are based on distinct statistical principles and therefore may yield partially divergent results. Our data confirm that while these tools generally point to overlapping sets of stable and unstable genes, discrepancies may still occur. This highlights the necessity of validating reference genes individually for each experimental system, taking into account specific cell types and experimental conditions. Genes that are stably expressed in one tissue or condition may not perform reliably in another. Furthermore, to ensure robust and unbiased normalization, it is strongly recommended to use multiple validated reference genes rather than relying on a single one. Combining reference genes for normalization can reduce variability and increase the accuracy of relative gene expression analyses.

## Figures and Tables

**Figure 1 ijms-26-06790-f001:**
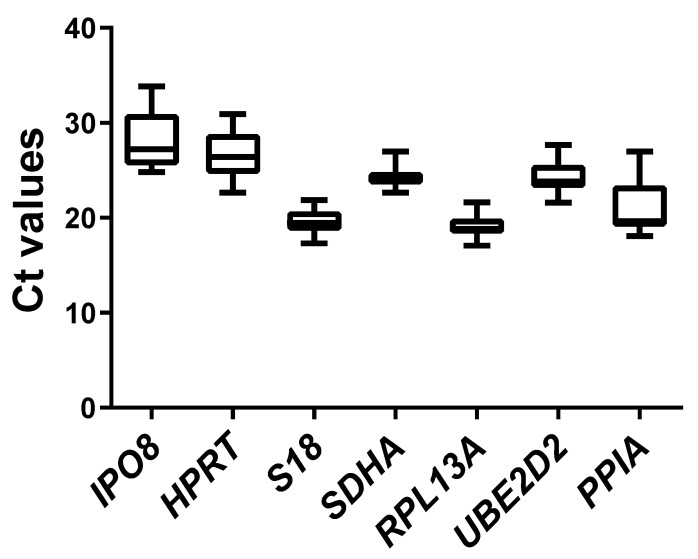
Ct values obtained after qPCR analysis for individual genes for all study groups. Box-and-whiskers plots representing the distribution of Ct values for a selected reference gene across all experimental conditions and treatment groups. Each box represents the interquartile range (25th–75th percentile), with the horizontal line indicating the median Ct value. Whiskers extend to the smallest and largest values.

**Figure 2 ijms-26-06790-f002:**
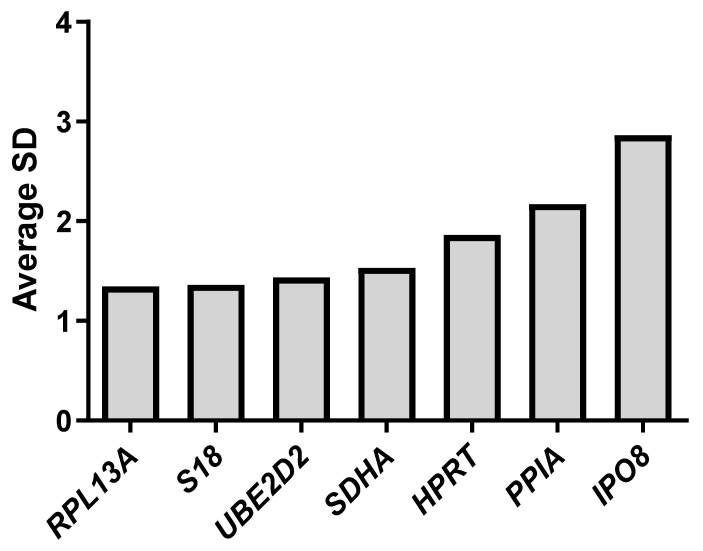
Mean standard deviations (SD) of ΔCt values for each gene. The highest variability was observed for *IPO8* and *PPIA*, whereas the lowest SD values were recorded for *RPL13A*, *S18*, and *UBE2D2*.

**Figure 3 ijms-26-06790-f003:**
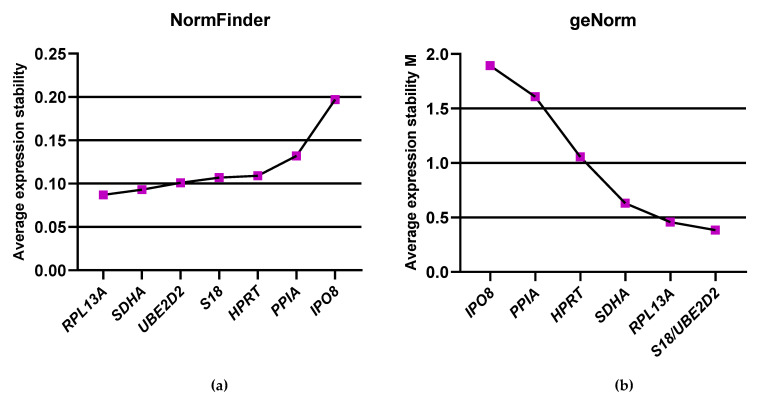
Ranking of reference genes by expression stability using NormFinder (**a**) and geNorm (**b**) algorithms. Both algorithms identify *IPO8* as the least suitable gene for qPCR data normalization. The most suitable reference gene pairs are *UBE2D2* and *S18* according to GeNorm and *RPL13A* and *SDHA* based on the NormFinder.

**Figure 4 ijms-26-06790-f004:**
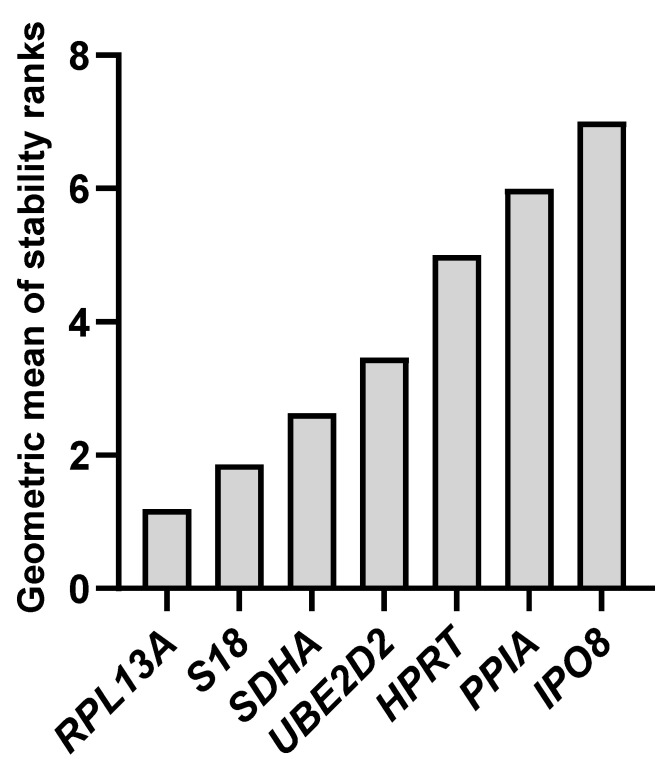
Analysis of reference gene stability by RefFinder. Classification of candidate reference genes based on stability values determined by geNorm, NormFinder, BestKeepr, and ΔCt methods, along with a recommended comprehensive ranking calculated using RefFinder algorithm. The results presented reflect the analysis performed across all experimental conditions: normoxia, hypoxia, and chemically induced hypoxia.

**Figure 5 ijms-26-06790-f005:**
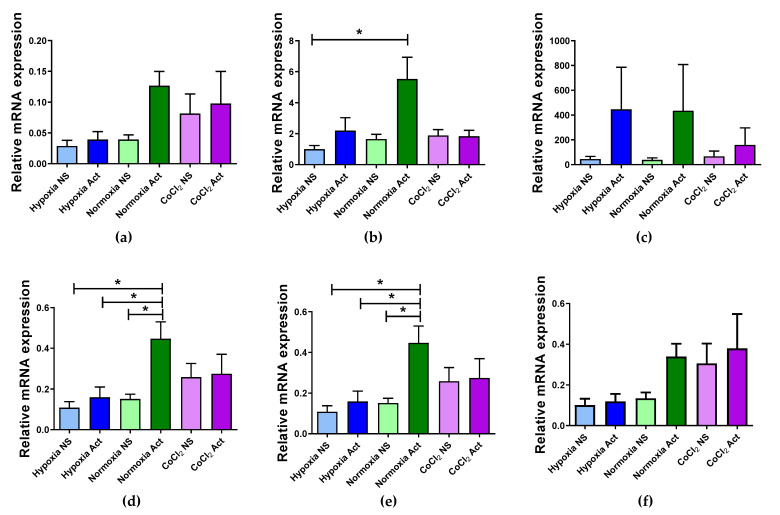
Relative expression of the *HIF1A* gene normalized to different reference genes. Expression levels were calculated by normalization to single reference genes: *RPL13A* (**a**), *SDHA* (**b**), and *IPO8* (**c**); and to multiple reference genes using either average (*RPL13A*/*S18*/*SDHA* (**d**)) or geometric mean (*RPL13A*/*S18*/*SDHA* (**e**); *UBE2D2*/*S18*/*RPL13A* (**f**)). Data represented as mean ± SEM, groups denoted with * are significantly different (*p* < 0.05, one-way ANOVA with Tukey’s post hoc test).

**Table 1 ijms-26-06790-t001:** References genes used in the study.

Abbreviation	Gene Name	NCBI Accession Number	Gene Function *
*HPRT*	Hypoxanthine Phosphoribosyltransferase 1	NM_000194	Involved in the purine salvage pathway
*S18*	Ribosomal Protein S18	NM_022551.3	Initiation and fidelity of translation as a core component of the small ribosomal subunit
*IPO8*	Importin 8	NM_001190995.2	Nuclear transport receptor mediates the proteins and ribonucleoproteins into the nucleus
*RPL13A*	Ribosomal Protein L13a	NM_001270491.2	Contributes to protein synthesis by participating in the assembly and function of ribosomes
*UBE2D2*	Ubiquitin Conjugating Enzyme E2 D2	NM_181838.2	Part of the ubiquitination system, acts as an E2 ubiquitin-conjugating enzyme and works with E3 ligases to attach ubiquitin to target proteins/regulates protein turnover, cell cycle progression, and responses to stress
*PPIA*	Peptidylprolyl Isomerase A	NM_001300981.2	Protein folding, intracellular signaling, inflammation, and viral infection processes
*SDHA*	Succinate Dehydrogenase Complex Flavoprotein Subunit A	NM_001294332.2	Catalyzes the oxidation of succinate to fumarate and transfers electrons to the electron transport chain

* information about gene functions was retrieved from the GeneCards database.

**Table 2 ijms-26-06790-t002:** Primer sequences, amplification efficiencies, and product sizes of seven selected reference genes.

Gene	Primer Sequence (5′-3′)	Efficiency [%]	R^2^	Slope	Amplicon Size (bp)	Source
*HPRT*	F: CCTGGCGTCGTGATTAGTGAT R: AGACGTTCAGTCCTGTCCATAA	113.5	0.992	−3.036	131	de novo
*S18*	F: TGGTCTGGACAACAAGCTCC R: GAAGTGACGCAGCCCTCTAT	103.93	0.996	−3.231	76	de novo
*IPO8*	F: GTGTAAGCTTCGTGAGGGC R: TGTGAGTTGCAGAAGACGGA	91.13	0.996	−3.555	108	de novo
*RPL13A*	F: AAAAGCGGATGGTGGTTCCT R: GCTGTCACTGCCTGGTACTT	98.32	0.999	−3.363	118	[21]
*UBE2D2*	F: ATTGAATGATCTGGCACGGG R: GTCATTTGGCCCCATTATTG	92.07	0.997	−3.528	100	[25]
*PPIA*	F:TGAGAACTTCATCCTAAAGCATAC R: CATCCAACCACTCAGTCTTG	113.6	0.999	−3.034	116	[26]
*SDHA*	F: TATATGGAAGGTCTCTGCGA R: GTGTTCTTTGCTCTTATGCG	99.77	0.987	−3.328	145	[26]

**Table 3 ijms-26-06790-t003:** Descriptive statistics of reference gene expression for all examined reference gene candidates by BestKeeper.

	IPO8	HPRT	S18	SDHA	RPL13A	UBE2D2	PPIA
**Geometric mean [CP]**	28.04	26.53	19.53	24.22	19.03	24.23	20.85
**Arithmetic mean [CP]**	28.19	26.64	19.57	24.24	19.07	24.28	20.99
**Minimum [CP]**	24.82	22.64	17.33	22.65	17.10	21.60	18.08
**Maximum [CP]**	33.86	30.90	21.89	26.99	21.63	27.67	26.99
**Standard deviation [±CP]**	2.76	1.95	1.07	0.83	0.97	1.35	2.19
**Coefficient variation [%CP]**	9.79	7.33	5.49	3.44	5.10	5.56	10.43
**Minimum [x-fold]**	−6.88	−24.38	−4.99	−2.95	−3.71	−4.98	−9.65
**Maximum [x-fold]**	32.68	35.98	5.61	6.76	5.79	8.16	155.29
**Standard deviation [±x-fold]**	5.22	3.22	1.90	1.65	1.79	2.25	3.71

**Table 4 ijms-26-06790-t004:** Classification of reference genes candidates based on stability values across all experimental conditions, as determined by the RefFinder algorithm.

Ranking Order (Better-Good-Average)							
**Conditions**	1	2	3	4	5	6	7
**Normoxia**	*RPL13A*	*UBE2D2*	*SDHA*	*S18*	*HPRT*	*PPIA*	*IPO8*
**Hypoxia**	*RPL13A*	*UBE2D2*	*SDHA*	*S18*	*HPRT*	*PPIA*	*IPO8*
**Chemical hypoxia**	*RPL13A*	*S18*	*HPRT*	*UBE2D2*	*SDHA*	*IPO8*	*PPIA*
**Recommended comprehensive ranking**	** *RPL13A* **	** *UBE2D2* **	** *S18* **	** *SDHA* **	** *HPRT* **	** *PPIA* **	** *IPO8* **

**Table 5 ijms-26-06790-t005:** Characteristics of healthy donors providing buffy coats.

Donor No.	Sex	Age	Blood Type	Rh (D)
**NDA**	Male	44	B	+
**NDB**	Male	54	A	−
**NDC**	Male	33	B	+
**NDD**	Male	33	0	+
**NDE**	Male	45	0	+

## Data Availability

The raw qPCR data (Ct values) generated and analyzed in this study are publicly available in the RepOD repository under the following DOI: https://doi.org/10.18150/UIYUCE (note: the link is pending activation). Additional datasets or materials are available from the corresponding author upon reasonable request.

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
