# Peer review of "Selection of Stable Reference Genes for Gene Expression Studies in Activated and Non-Activated PBMCs Under Normoxic and Hypoxic Conditions"

_ijms, 2025, doi:10.3390/ijms26146790_

Round 1

Reviewer 1 Report

Comments and Suggestions for Authors

Manuscript entitled “Selection of Stable Reference Genes for Gene Expression Studies in Activated and Non‑Activated PBMCs under Normoxic and Hypoxic Conditions” by Artur Wardaszka, Anna Smolarska, Piotr Bednarczyk & Joanna Bujak

The study addresses a genuine translational bottleneck: the lack of rigorously validated reference (house‑keeping) genes for quantitative PCR analyses in peripheral blood mononuclear cells (PBMCs) exposed to hypoxia, a hallmark of the tumour tumour micro‑environment. By systematically comparing seven candidate genes with four complementary algorithms (ΔCt, geNorm, NormFinder and BestKeeper) and experimentally validating the winning set on HIF‑1α expression, the authors generate a valuable methodological resource for immunology and oncology laboratories. Their thorough primer‑efficiency determination, multi‑algorithm ranking and use of both physical (1 % Oâ‚‚) and chemical (CoClâ‚‚) hypoxia represent clear strengths. 

Comments:

1. Introduction must be substantially condensed and refocused. The current Introduction spends >600 words reviewing general cancer immunotherapy and basic hypoxia biology before mentioning reference‑gene validation; key background for PBMC qPCR normalisation is therefore diluted. Lines 40 – 73 could be shortened and citations rationalised.

2. Define and standardise first use of all abbreviations. PBMC, TME, HIF, CAR‑T, CV, ΔCt, etc. appear before definition (e.g. PBMC first occurs in line 56 of the Introduction without expansion) . 

3. The Methods state that “healthy human donor PBMCs were isolated from buffy coats” but do not report the number, sex, age range or inclusion/exclusion criteria of donors, nor ethical approval details. Clarify donor characteristics and sample size justification; inter‑donor variation is particularly relevant as Figure S2/S3 reveal donor‑specific Ct dispersion for IPO8 and PPIA. 

4. The Discussion cites a handful of prior PBMC or T‑cell papers but does not quantitatively compare stability rankings. Integrate direct comparisons (e.g., why RPL13A ranked poorly in influenza‑infected PBMCs yet excelled here; reconcile with IPO8 stability in lung tissue). 

5. Study limitations require explicit acknowledgement.

6. Language and typographical edits.

 Recurrent minor errors (e.g., “realtively”, “conditon”, “transription”) should be corrected. The manuscript would benefit from professional language editing for flow and concision

Author Response

We sincerely thank you for your valuable comments, insightful questions, and constructive suggestions, which helped us to improve the quality and clarity of the manuscript. 

Comment 1: The Introduction must be substantially condensed and refocused. The current Introduction spends >600 words reviewing general cancer immunotherapy and basic hypoxia biology before mentioning reference gene validation; key background for PBMC qPCR normalisation is therefore diluted. Lines 40 – 73 could be shortened and citations rationalised. 

Response 1:  

Thank you for your thoughtful comments. We applied the mentioned changes to the Introduction. We shortened the section of the manuscript related to cancer immunotherapy and hypoxia biology as requested.  

We were, however, asked by the second reviewer to include in the Introduction an explanation of the different types of immunotherapy and which of them are most sensitive to hypoxia, along with a description of how hypoxia affects both neoplastic cells and cells of the tumor microenvironment (TME) (lines 45–57, 62-65). Additionally, we addressed the distinction between physiological and chemically induced hypoxia (lines 76-80). These modifications were introduced in a manner that did not substantially increase the length of the Introduction section. 

Comment 2: Define and standardise the first use of all abbreviations. PBMC, TME, HIF, CART, CV, ΔCt, etc. appear before definition (e.g., PBMC first occurs in line 56 of the Introduction without expansion).  

Response 2:  

We have revised the manuscript to define and standardise all abbreviations at their first occurrence, including PBMC, TME, HIF, CART, CV, and ΔCt, as requested. 

Comment 3: The Methods state that “healthy human donor PBMCs were isolated from buffy coats” but do not report the number, sex, age range, or inclusion/exclusion criteria of donors, nor ethical approval details. Clarify donor characteristics and sample size justification; inter‑donor variation is particularly relevant as Figure S2/S3 reveals donor‑specific Ct dispersion for IPO8 and PPIA. 

Response 3: 

Thank you for these valuable comments. Information regarding donor characteristics and sample origin has been added to the revised manuscript. A summary Table 5, including the number of donors, sex distribution, and age range, is now provided in the Materials and Methods section (line 567). 

Regarding sample size, PBMCs were isolated from buffy coats obtained from healthy adult donors (n=5, age range: 33-54). The buffy coats were provided by the Regional Blood Donation and Blood Treatment Center of the Ministry of Interior and Administration in Warsaw, Poland. All donors provided informed consent at the time of blood donation, in accordance with the institution’s standard procedures.  

All donors were male, which reflects that the majority of voluntary donors in our local blood center are men. Additionally, we aimed to minimise variability related to hormonal fluctuations, which are more prominent in females due to the menstrual cycle and can affect immune cell activity. We acknowledge that sex-related differences in gene expression and immune responses are relevant and warrant further investigation in future studies with a more balanced donor population. We also added this information in the discussion section as a limitation of the study (lines 280- 285 ).  

To reduce technical variability, all samples were processed in parallel to minimize plate-to-plate variation. Samples from donors showing signs of pre-existing activation were excluded from the analysis, as they may have originated from non-healthy individuals and could obscure the interpretation of experimentally induced activation effects.  

We acknowledge that inter-donor variability is an inherent feature of studies using primary human cells. This biological diversity, while sometimes challenging, reflects the true complexity of the immune system and underlines the importance of validating reference genes in a donor-inclusive manner. Nevertheless, the NormFinder algorithm explicitly accounts for inter- and intra-donor variation when assessing gene stability. The observed Ct dispersion across donors (as shown in Figure S3) further supports the necessity of such normalization strategies. 

Due to the nature of the material (primary PBMCs), we have no experimental control over the natural inter-individual variation, but we see it as a relevant aspect that highlights the practical importance of our findings in the context of human biological diversity. We highlight the inter-donor variation in the Discussion sections (lines 395 - 406). 

Comment 4: The Discussion cites a handful of prior PBMC or T cell papers but does not quantitatively compare stability rankings. Integrate direct comparisons (e.g., why RPL13A ranked poorly in influenza-infected PBMCs yet excelled here; reconcile with IPO8 stability in lung tissue).  

Response 4: Thank you for your valuable comments. We appreciate your suggestion to include more direct and quantitative comparisons of reference gene stability with previously published studies. However, such comparisons are inherently challenging due to substantial differences in experimental design, biological material, and conditions across studies. 

For example, many published PBMC datasets involve cells from patients with specific conditions such as asthma, autoimmune diseases, or viral infections (e.g., influenza), all of which can markedly influence gene expression profiles and reference gene stability. In contrast, our study was conducted on PBMCs isolated from healthy male donors and subjected to controlled hypoxic conditions. These biological and experimental discrepancies may explain the differential behavior of reference genes — for instance, RPL13A ranked poorly in influenza-infected PBMCs and T cells (as RPL13A expression is known to be affected by viral infections), but performed better in CD3/CD28-activated T cells.  

Furthermore, PBMCs are a heterogeneous population comprising various immune cell subsets (e.g., T cells, B cells, monocytes, NK cells), and their relative composition can vary significantly between individuals. This inter-donor variability contributes to the observed differences in gene expression. In contrast, many previously published reference gene studies, such as those involving IPO8, were performed in clonal cell lines, which represent more uniform and reproducible systems. For example, IPO8 was identified as a stable reference gene in clinical lung tissue samples; however, in the same study, it was among the least stable genes in lung cell lines cultured in vitro. We included this specific case in the revised Discussion to illustrate how gene stability can be highly context-dependent, depending not only on tissue type but also on whether the samples are in vivo or in vitro. 

To aid in the interpretation of our data, we have also underlined the commonly accepted stability thresholds for each algorithm. In both NormFinder and geNorm, lower M values indicate greater stability; generally, an M value below 0.15 in NormFinder and below 1.5 in geNorm is considered acceptable. While these thresholds are indeed arbitrary and can vary slightly between publications, all of our top-performing candidate genes fall well within these ranges, supporting their appropriateness for accurate normalization in our experimental context. 

We have added these clarifications to the revised Discussion (lines 395 – 406, 438 - 458), including specific numeric values where appropriate.  

Comment 5: Study limitations require explicit acknowledgement. 

Response 5: The limitations of our study—including the small sample size and the use of only male donors—have now been explicitly addressed in the revised Discussion section [lines 280 - 285]. 

Comment 6: Language and typographical edits. 

Recurrent minor errors (e.g., “realtively”, “conditon”, “transription”) should be corrected. The manuscript would benefit from professional language editing for flow and concision 

Response 6: 

The manuscript has been carefully revised to correct typographical errors.  

Reviewer 2 Report

Comments and Suggestions for Authors

This study analyzed the stability of reference genes under hypoxia in PBMC. Neoplasia is often associated with angiogenesis but also with hipoxia within the tumor. Therefore, reference genes (I understand hosekeeping genes) have to be properly selected.
Comments:

(1) Lines 41-48. There are many types of immune theraphy. Could you please describe in more detail the types of immuno-oncology therapies and the ones that are sensitive to hypoxia? For example, how hypoxia affect the neoplastic cells and the TME cells and the relationship with immuno-oncology therapies.

(2) The study targets the PBMC. Therefore, T, B, NK cells, and monocytes. Please note the granulocytes, erythrocytes and platelets are exluded in the analysis. Please not that these cells are also the origin of hematological neoplasia. Please add comment if necessary in the text.

(3) Line 54. Which cells upregulate HIF in TME under hypoxia?

(4) Line 71. What is the difference between O2 and chemical hypoxia?

(5) Line 73. Why these genes were selected? What was the criteria? I know you mention literature review, but could you please explain in detail the resone?

(6) Line 77. Could you please explain the methods and the reasoning of using "geNorm, NormFinder, BestKeeper, and the comparative ΔCt method"?

(7) Table 1. Regarding column "gene function". How was this data searched?

(8) Line 96. Can the efficiency be above 100%?

(9) Please confirm that all primer sequences are correct.

(10) Table 2. How were the primer sequences designed "de novo"?

(11) All all PBMC stimulated with CD3/CD28?

(12) In Figure 1. Are all PCR products the same amount?

(13) As I understand, up to line 209, the analysis is made under normal conditions without hypoxia. Is this correct?

(14) Why normoxia act has higher expression of HIF1A than hypoxia?

(15) Are the results stratified according to the subtype of PBMC?

(16) If the samples were neoplastic, would the expression of thesese genes be also changed? Are these genes frequenty altered (copy number, mutation, methylation, etc.) during neoplasia?

Author Response

We sincerely thank you for your valuable comments, insightful questions, and constructive suggestions, which helped us to improve the quality and clarity of the manuscript.

Comment 1: Lines 41-48. There are many types of immune therapy. Could you please describe in more detail the types of immuno-oncology therapies and the ones that are sensitive to hypoxia? For example, how hypoxia affect the neoplastic cells and the TME cells and the relationship with immuno-oncology therapies. 

Response 1: 

 Thank you for your valuable and thoughtful comments. In the Introduction section (lines 50-59), we have included information regarding the types of immunotherapies most sensitive to hypoxia, as well as a brief explanation that hypoxia promotes the survival of neoplastic cells and enhances their resistance to immune effector cells present in the TME.  

As we were asked by Reviewer 1 to shorten the Introduction, a detailed discussion of specific immunotherapy types was not included, as the primary focus of this manuscript is the selection of suitable reference genes for PBMC under hypoxic conditions. Nevertheless, we fully acknowledge the importance of this topic. 

Comment 2: The study targets the PBMC. Therefore, T, B, NK cells, and monocytes. Please note the granulocytes, erythrocytes and platelets are exluded in the analysis. Please not that these cells are also the origin of hematological neoplasia. Please add comment if necessary in the text. 

Response 2: A corresponding note has now been added to the revised version of the manuscript [Materials and methods section, lines 568 -570]. 

Comment 3: Line 54. Which cells upregulate HIF in TME under hypoxia? 

Response 3: Under hypoxic conditions in the TME, HIFs are upregulated in a wide range of cell types, including both neoplastic and stromal cells. Specifically, cancer cells stabilize HIFs to support survival, angiogenesis, and metabolic adaptation. In addition, various immune cells within the TME, such as T lymphocytes, tumor-associated macrophages, dendritic cells, and myeloid-derived suppressor cells, also upregulate HIFs in response to low oxygen availability. This upregulation modulates their phenotype and function, often contributing to immunosuppression and tumor immune escape.  
As suggested, this information has been added in the Introduction section (lines 62-65). 

Comment 4: Line 71. What is the difference between O2 and chemical hypoxia? 
Response 4:  

We acknowledge the importance of distinguishing between hypoxia and chemical hypoxia, as their underlying mechanisms differ significantly. Physiological/physical hypoxia results from decreased partial pressure of oxygen in the environment, leading to decreased oxygen saturation of hemoglobin and reduced oxygen delivery to tissues. In contrast, chemical hypoxia induced by cobalt chloride does not involve a drop in oxygen levels. Instead, CoCl2 mimics hypoxic conditions by stabilizing HIF1α under normoxic conditions. Stabilized HIF1α accumulates in the cell and activates hypoxia-responsive genes, thereby functionally stimulating a hypoxic cellular response despite normal oxygen availability.   

We have updated the manuscript accordingly to clarify these mechanisms in the Introduction section (lines 76-80).   

Comment 5: Line 73. Why these genes were selected? What was the criteria? I know you mention literature review, but could you please explain in detail the resone? 

Response 5: We appreciate your thoughtful question concerning the criteria for selecting candidate reference genes. The choice of candidate reference genes was guided not only by their frequent use in gene expression studies but also by their reported stability under stress-related and hypoxia-mimicking conditions. We selected genes involved in distinct and unrelated cellular processes – including ribosomal function, mitochondrial activity, protein transport, and enzymatic metabolism – to reduce the risk of shared regulatory responses under hypoxic stress. Furthermore, our selection was informed by previous studies identifying stable reference genes in PBMCs and T cells under various conditions, as well as by research conducted on livestock species naturally adapted to high-altitude hypoxic environments, in which certain reference genes demonstrated consistent stability. These findings provided additional support for the inclusion of selected genes in our analysis.  

The appropriate references have been included in the introduction section (lines 80-82).  

Comment 6: Line 77. Could you please explain the methods and the reasoning of using "geNorm, NormFinder, BestKeeper, and the comparative ΔCt method"? 

Response 6:  

Thank you for your question. The methods geNorm, NormFinder, BestKeeper, and the comparative ΔCt method are commonly used approaches for the selection of reference genes in qPCR studies. Each of these tools relies on slightly different statistical algorithms to assess gene expression stability, which allows for a more comprehensive evaluation. Using a combination of these tools provides a more robust and comprehensive evaluation. This multi-method strategy is particularly important when working with primary cells from different donors, where biological variability may influence gene stability. The specific differences between these methods are explained in detail in lines 315-354 of the Discussion section. 

Comment 7: Table 1. Regarding column "gene function". How was this data searched? 

Response 7: Thank you for the comment regarding the source of the gene function data presented in Table 1. The information in the “gene function” column was obtained from the GeneCards database (https://www.genecards.org), a comprehensive resource that compiles gene-related data from multiple databases, including UniProt, HGNC, Entrez Gene, and scientific literature. A reference to GeneCards has been added to the manuscript (the Table 1. legend) to clearly indicate the source of these annotations.     

Comment: 8 Line 96. Can the efficiency be above 100%? 

Response 8: Primer efficiency above 100% can occur due to several reasons, such as technical variations, such as pipetting inaccuracies, baseline correction, amplification of non-specific products or primer-dimers, or presence of inhibitors in standards. However, in our case, the efficiencies were consistently close to 100% and within the acceptable range (a good range of efficiency is around 90-110%), indicating reliable performance. We have reviewed all standard curves and melting curves to ensure the specificity and validity of the results. 

Comment 9: Please confirm that all primer sequences are correct. 

Response 9: We have carefully checked all primer sequences and confirm that they are correct.  

Comment 10: Table 2. How were the primer sequences designed "de novo"? 

Response 10: We appreciate your interest in this aspect of the study and welcome the opportunity to explain it in more detail. The de novo-designed primer sequence was generated as follows: mRNA sequence in FASTA format for the relevant genes (HPRT, S18, IPO8) was retrieved from the NCBI database. Primer-BLAST, an online design tool, was then used to select appropriate primer pairs. The parameters defined during primer selection included a PCR product size of 70-200 bp and a primer melting temperature (Tm) with an optimal value of 55 °C and a maximum Tm difference between primer pairs of 3 °C. During primer design, optimal GC content (40% < GC < 60%) was taken into consideration, and care was taken to avoid G/C repeats at the 5' and 3' ends. Additionally, parameters such as self-complementarity and 3' self-complementarity were evaluated to minimize the risk of secondary structure formation. Additionally, all primer sequences were verified to ensure no significant similarity to non-target sequences, thereby minimizing the risk of off-target amplification and ensuring the specificity of the qPCR reactions. 

Comment 11: All all PBMC stimulated with CD3/CD28? 

Response 11: Thank you for your insightful question. Not all PBMCs were stimulated with CD3/CD28, as this stimulation specifically targets T lymphocytes, which express the CD3 and CD28 surface receptors. However, since lymphocytes constitute a major proportion of the PBMC population, the majority of cells in our cultures were indeed responsive to CD3/CD28 activation. This cellular composition was confirmed by cell-type distribution analysis using a hemocytometer. Additionally, it is worth noting that although monocytes and other non-T-cell populations within PBMCs are not directly activated by CD3/CD28, they may still be indirectly affected by cytokines and signaling molecules released by activated T cells.  

Comment 12: In Figure 1. Are all PCR products the same amount? 

Response 12: Figure 1 presents the Ct values obtained for individual genes under each experimental condition. These values reflect the relative abundance of each transcript in the total mRNA pool, and as shown, not all genes are expressed at the same levels—some have lower Ct values, indicating higher expression, while others are less abundant. Thus, the PCR products (cDNA amplicons) differ in amount, consistent with natural variation in transcript levels between genes. 

Importantly, we used equal amounts of total mRNA for reverse transcription across all samples, ensuring that each qPCR reaction was performed with cDNA synthesized from the same starting quantity of RNA. This approach allowed us to compare gene expression across conditions in a standardized manner. 

Comment 13: As I understand, up to line 209, the analysis is made under normal conditions without hypoxia. Is this correct? 

Response 13: Thank you for your question and the opportunity to provide a more detailed explanation. Stability analysis was conducted across all experimental conditions – normoxia, hypoxia, and chemical hypoxia – using four established algorithms: NormFinder, GeNorm, BestKeeper, and the ΔCt method. However, in accordance with your valuable suggestion, we have added this information beneath Figure 4 (line 220 - 221).  

Comment 14: Why normoxia act has higher expression of HIF1A than hypoxia? 

Response 14: Thank you for your question. The unexpectedly higher HIF1α expression observed under normoxic conditions compared to hypoxia in PBMC, composed mainly of lymphocytes, may be attributed to a combination of regulatory mechanisms and cellular metabolic states. Lymphocytes, upon activation, undergo metabolic reprogramming toward glycolysis, a process associated with HIF1α transcriptional activity. Although HIF1α is classically stabilized at the protein level under hypoxia, its expression is also upregulated in activated lymphocytes under normoxia to support glycolytic pathways. Additionally, HIF1α influences T cell differentiation through the mTOR signaling pathway (https://doi.org/10.1084/jem.20110278). This study was also cited in the Discussion section, where importance of HIF1α in T cell physiology is mentioned (lines 536-546). 

Comment 15: Are the results stratified according to the subtype of PBMC? 

Response 15: Our study was performed on total peripheral blood mononuclear cells (PBMCs), without separating them into specific subpopulations. Therefore, the results presented are based on the mixed PBMC population, as isolated from the whole blood. While it is true that T lymphocytes typically constitute the majority of PBMCs, we did not perform additional stratification or isolation of specific cell types such as T cells, B cells, monocytes, or NK cells in this study. 

We acknowledge that further studies focusing on individual PBMC subtypes could provide deeper insight into gene expression differences within specific immune cell populations. 

Comment 16: If the samples were neoplastic, would the expression of thesese genes be also changed? Are these genes frequenty altered (copy number, mutation, methylation, etc.) during neoplasia? 

Response 16:  

Given that our study focused on non-neoplastic samples, we did not assess these genes under neoplastic conditions. However, we agree that their stability should be carefully evaluated if applied in cancer-related studies. Neoplastic cells often exhibit profoundly altered regulatory networks and can upregulate or downregulate genes that are typically considered stable in healthy tissues. Therefore, genes like IPO8, RPL13A, UBE2D2, SDHA, HPRT1, PPIA, and 18S, although frequently used as reference genes, should be individually assessed for expression stability in neoplastic conditions before being applied as normalizers in gene expression studies involving tumors. 

We have noted in the Discussion section (lines 504 - 509) and conclusions (lines 665 – 676) emphasizing that the stability of reference genes should always be carefully validated for each specific experimental condition, rather than relying solely on their prior use in the literature. Housekeeping genes may exhibit stable expression in many contexts, but this cannot be assumed universally. 

Round 2

Reviewer 1 Report

Comments and Suggestions for Authors

The authors have adequately addressed all of my concerns.